# Depth Anywhere: Enhancing 360 Monocular Depth Estimation via Perspective Distillation and Unlabeled Data Augmentation

**Ning-Hsu Wang**
albert.nhwang@gmail.com

**Yu-Lun Liu**
Department of Computer Science
National Yang Ming Chiao Tung University
yulunliu@cs.nycu.edu.tw

## Abstract

Accurately estimating depth in 360-degree imagery is crucial for virtual reality, autonomous navigation, and immersive media applications. Existing depth estimation methods designed for perspective-view imagery fail when applied to 360-degree images due to different camera projections and distortions, whereas 360-degree methods perform inferior due to the lack of labeled data pairs. We propose a new depth estimation framework that utilizes unlabeled 360-degree data effectively. Our approach uses state-of-the-art perspective depth estimation models as teacher models to generate pseudo labels through a six-face cube projection technique, enabling efficient labeling of depth in 360-degree images. This method leverages the increasing availability of large datasets. Our approach includes two main stages: offline mask generation for invalid regions and an online semi-supervised joint training regime. We tested our approach on benchmark datasets such as Matterport3D and Stanford2D3D, showing significant improvements in depth estimation accuracy, particularly in zero-shot scenarios. Our proposed training pipeline can enhance any 360 monocular depth estimator and demonstrates effective knowledge transfer across different camera projections and data types. See our project page for results: albert100121.github.io/Depth-Anywhere.

## 1 Introduction

In recent years, the field of computer vision has seen a surge in research focused on addressing the challenges associated with processing 360-degree images. The widespread use of panoramic imagery across various domains, such as virtual reality, autonomous navigation, and immersive media, has underscored the need for accurate depth estimation techniques tailored specifically for 360-degree images. However, existing depth estimation methods developed for perspective-view images encounter significant difficulties when applied directly to 360-degree data due to differences in camera projection and distortion. While many methods aim to address depth estimation for this camera projection, they often struggle due to the limited availability of labeled datasets.

To overcome these challenges, this paper presents a novel approach for training state-of-the-art (SOTA) depth estimation models on 360-degree imagery. With the recent significant increase in the amount of available data, the importance of both data quantity and quality has become evident. Research efforts on perspective perceptual models have increasingly focused on augmenting the volume of data and developing foundation models that generalize across various types of data. Our method leverages SOTA perspective depth estimation foundation models as teacher models and generates pseudo labels for unlabeled 360-degree images using a six-face cube projection approach. By doing so, we efficiently address the challenge of labeling depth in 360-degree imagery by leveraging perspective models and large amounts of unlabeled data.

38th Conference on Neural Information Processing Systems (NeurIPS 2024).

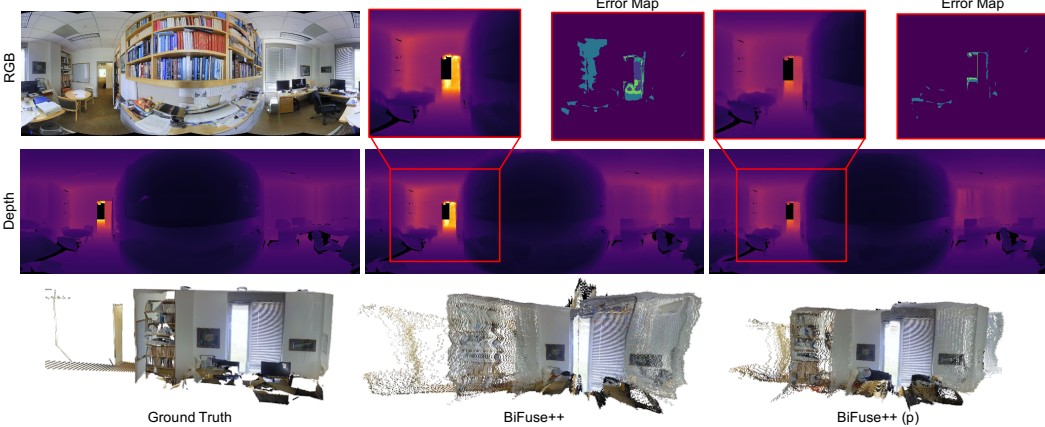

Figure 1: **Our proposed training pipeline improves existing 360 monocular depth estimators.** This figure demonstrated the improvement of our proposed training pipeline tested on the Stanford2D3D [2] dataset in a zero-shot setting.

Our approach consists of two key stages: offline mask generation and online joint training. During the offline stage, we employ a combination of detection and segmentation models to generate masks for invalid regions, such as sky and watermarks in unlabeled data. Subsequently, in the online stage, we adopt a semi-supervised learning strategy, loading half of the batch with labeled data and the other half with pseudo-labeled data. Through joint training with both labeled and pseudo-labeled data, our method achieves robust depth estimation performance on 360-degree imagery.

To validate the effectiveness of our approach, we conduct extensive experiments on benchmark datasets such as Matterport3D and Stanford2D3D. Our method demonstrates significant improvements in depth estimation accuracy, particularly in zero-shot scenarios where models are trained on one dataset and evaluated on another. Furthermore, we demonstrate the efficacy of our training techniques with different SOTA 360-degree depth models and various unlabeled datasets, showcasing the versatility and effectiveness of our approach in addressing the unique challenges posed by 360-degree imagery.

Our contributions can be summarized as follows:

- We propose a novel training technique for 360-degree imagery that harnesses the power of unlabeled data through the distillation of perspective foundation models.

- We introduce an online data augmentation method that effectively bridges knowledge distillation across different camera projections.

- Our proposed training techniques significantly benefit and inspire future research on 360-degree imagery by showcasing the interchangeability of state-of-the-art (SOTA) 360 models, perspective teacher models, and unlabeled datasets. This enables better results even as new SOTA techniques emerge in the future.

## 2 Related Work

**360 monocular depth.** Depth estimation for 360-degree images presents unique challenges due to the equirectangular projection and inherent distortion. Various approaches have been explored to address these issues:

- *Directly Apply*: Some methods directly apply monocular depth estimation techniques to 360-degree imagery. OmniDepth [69] leverages spherical geometry and incorporates Sph-Conv [42] to improve depth prediction with distortion. [70, 49] use spherical coordinates to overcome distortion with extra information. [12, 17] leverage other ground truth supervisions to assist on depth estimation. SliceNet [31] and ACDNet [68] propose advanced network architectures tailored for omnidirectional images. EGFormer [64] and HiMODE [18] introduce a transformer-based model that captures global context efficiently, while [45, 44]

focuses on integrating geometric priors into the learning process. [13] proposed to generate large-scale datasets with SfM and MVS, then apply to test-time training.

- *Cube*: Other approaches use cube map projections to mitigate distortion effects. 360-SelfNet [46] is the first work to self-supervised 360 depth estimation leveraging cube-padding [9]. BiFuse [47] and its improved version BiFuse++ [48] are two-branch architectures that utilize cube maps and equirectangular projections. UniFuse [16] combines equirectangular and cube map projections and simplifies the architecture. [3] combines two-branch techniques with transformer network.

- *Tangent Image*: Tangent image projections are also popular. [37, 23, 30] convert equirectangular images into a series of tangent images, which are then processed using conventional depth estimation networks. PanoFormer [40] employs a transformer-based architecture to handle tangent images, while SphereNet [11] and HRDFuse [1] enhance depth prediction by collaboratively learning from multiple projections.

**360 other works** Beyond depth estimation, 360-degree imagery has been applied to depth completion tasks as follows [26, 8, 32, 57, 58, 15]. Other methods, such as [25] and [43] focus on the projection between camera models, while the former projects pinhole camera model images into a large field of view, whereas the latter transforms convolution kernels.

**Unlabeled / Pseudo labeled data.** Utilizing unlabeled or pseudo-labeled data has become a significant trend to mitigate the limitations of labeled data scarcity. Techniques like [22, 71, 41, 56] leverage large amounts of unlabeled data to improve model performance through semi-supervised learning. In the context of 360-degree depth estimation, our approach generates pseudo labels from pre-trained perspective models, which are then used to train 360-degree depth models effectively.

**Zero-shot methods.** Zero-shot learning methods aim to generalize to new domains without additional training data. [7, 54] target this directly with increasing training data., MiDaS [35, 5, 34] and Depth Anything [59] are notable for their robust monocular depth estimation across diverse datasets leveraging affine-invariant loss. [61] takes a step further to investigate zero-shot on metric depth. Marigold [19] leverages diffusion models with image conditioning and up-to-scale relative depth denoising to generate detailed depth maps. ZoeDepth [4] further these advancements by incorporating scale awareness and domain adaptation. [14, 50] leverage camera model information to adapt cross-domain depth estimation.

**Foundation models.** Foundation models have revolutionized various fields in AI, including natural language processing and image-text alignment. In computer vision, models like CLIP [33] demonstrate exceptional generalization capabilities. [28] proposed a foundation visual encoder for downstream tasks such as segmentation, detection, depth estimation, etc. [20] proposed a model that can cut out masks for any objects. Our work leverages a pre-trained perspective depth estimation foundation model [59] as a teacher model to generate pseudo labels for 360-degree images, enhancing depth estimation by utilizing the vast knowledge embedded in these foundation models.

## 3 Methods

In this work, we propose a novel training approach for 360-degree monocular depth estimation models. Our method leverages a perspective depth estimation model as a teacher and generates pseudo labels for unlabeled 360-degree images using a 6-face cube projection. Figure 2 illustrates our training pipeline, incorporating the use of Segment Anything to mask out sky and watermark regions in unlabeled data during the offline stage. Subsequently, we conduct joint training using both labeled and unlabeled data, allocating half of the batch to each. The joint training avoids limiting performance by teacher model. The unlabeled data is supervised using pseudo labels generated by Depth Anything, a state-of-the-art perspective monocular depth foundation model. With the benefit of our teacher model, the 360-degree depth model demonstrates an observable improvement on the zero-shot dataset, as shown in Figure 1.

### 3.1 Unleashing the Power of Unlabel 360 data

**Dataset statistics.** 360-degree data has become increasingly available in recent years. However, compared to perspective-view depth datasets, labeling depth ground truths for 360-degree data presents greater challenges. Consequently, the availability of labeled datasets for 360-degree data is considerably smaller than that of perspective datasets.

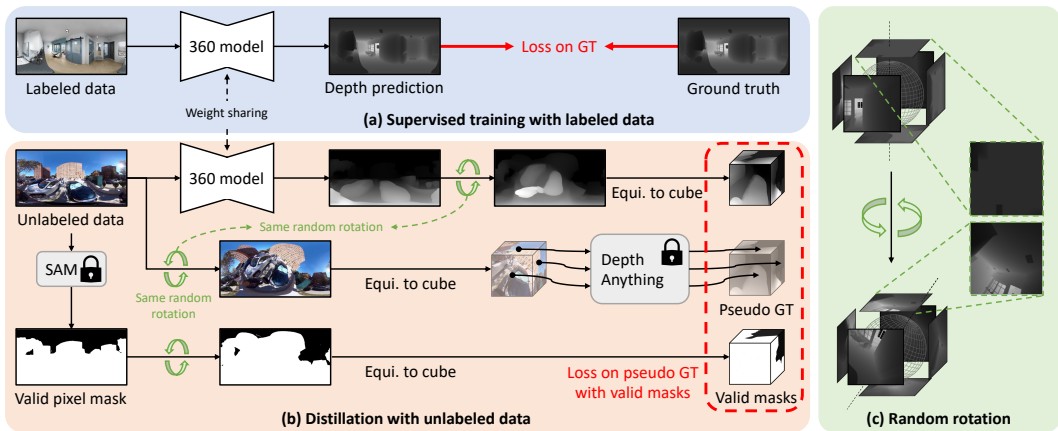

Figure 2: **Training Pipeline.** Our proposed training pipeline involves joint training on both labeled 360 data with ground truth and unlabeled 360 data. (a) For labeled data, we train our 360 depth model with the loss between depth prediction and ground truth. (b) For unlabeled data, we propose to distill knowledge from a pre-trained perspective-view monocular depth estimator. In this paper, we use Depth Anything [59] to generate pseudo ground truth for training. However, more advanced techniques could be applied. These perspective-view monocular depth estimators fail to produce reasonable equirectangular depth as a domain gap exists. Therefore, we distill knowledge by inferring six perspective cube faces and passing them through perspective-view monocular depth estimators. To ensure stable and effective training, we propose generating a valid pixel mask with Segment Anything [20] while calculating loss. (c) Furthermore, we augment random rotation on RGB before passing it into Depth Anything, as well as on predictions from the 360 depth model.

Table 1: **360 monocular depth estimation lacks a large amount of training data.** The number of images in 360-degree monocular depth estimation datasets alongside perspective depth datasets from the Depth Anything methodology.

| | Perspective | | | Equirectangular | | | |
|---|---|---|---|---|---|---|---|
| Labeled | 1.5M | Unlabeled | 62 M | Labeled | 34K | Unlabeled | 344K |

Table 1 presents the data quantities available in some of the most popular 360-degree datasets, including Matterport3D [6], Stanford2D3D [2], and Structured3D [65]. Additionally, we list a multi-modal dataset, SpatialAudioGen [29], which consists of unlabeled 360-degree data used in our experiments. Notably, the amount of labeled and unlabeled data used in the perspective foundation model, Depth Anything [59], is significantly larger, with 1.5 million labeled images [24, 52, 10, 60, 55, 51] and 62 million unlabeled images [38, 63, 53, 62, 39, 21, 66, 20], making the amount in 360-degree datasets approximately 170 times smaller.

**Data cleaning and valid pixel mask generation**    Unlabeled data often contains invalid pixels in regions such as the sky and watermark, leading to unstable training or undesired convergence. To address this issue, we applied the GroundingSAM [36] method to mask out the invalid regions. This approach utilizes Grounded DINOv2 [27] to detect problematic regions and applies the Segment Anything [20] model to mask out the invalid pixels by segmenting within the bounding box. While Depth Anything [59] also employs a pre-trained segmentation model, DINOv2, to select sky regions. Brand logos and watermarks frequently appear after fisheye camera stitching. Therefore, additional labels are applied to enhance the robustness of our training process. We also remove all images with less than 20 percent of valid pixels to stablize our training progress.

**Perspective foundation models (teacher models).**    To tackle the challenges posed by limited data and labeling difficulties in 360-degree datasets, we leverage a large amount of unlabeled data alongside state-of-the-art perspective depth foundation models. Due to significant differences in camera projection and distortion, directly applying perspective models to 360-degree data often yields inferior results. Previous works have explored various methods of projection for converting equirectangular to perspective depth, as stated in Section 2. Among these, cube projection and tangent

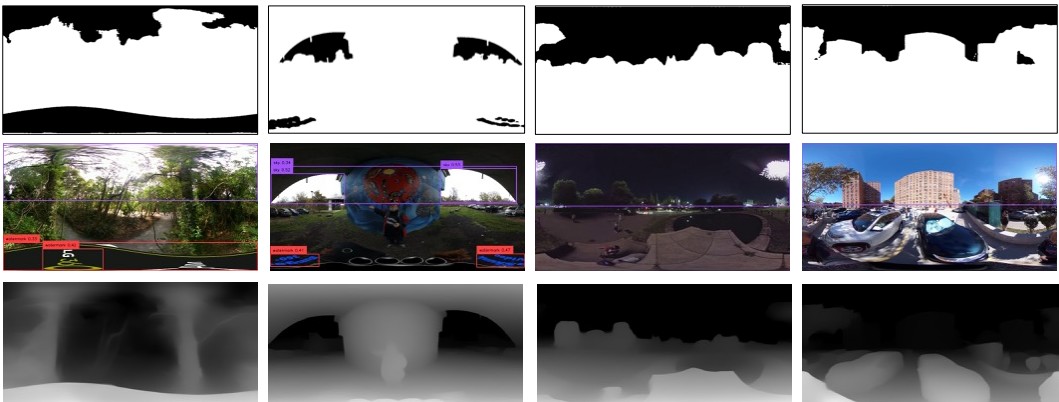

Figure 3: **Valid Pixel Masking.** We used Grounded-Segment-Anything [36] to mask out invalid pixels based on two text prompts: "sky" and "watermark." These regions lack depth sensor ground truth labels in all previous datasets. Unlike Depth Anything [59], which sets sky regions as 0 disparity, we follow ground truth training to ignore these regions during training for two reasons: (1) segmentation may misclassify and set other regions as zero, leading to noisy labeling, and (2) watermarks are post-processing regions that lack geometrical meaning.

projection are the most common techniques. We selected cube projection to ensure a larger field of view for each patch, enabling better observation of relative distances between pixels or objects during the inference of the perspective foundation model and enhancing knowledge distillation. The comparison table can be find in the supplementary material.

In our approach, we apply projection to unlabeled 360-degree data and then run Depth Anything on these projected patches of perspective images to generate pseudo-labels. We explore two directions for pseudo-label supervision: projecting the patch to equirectangular and computing in the 360-degree domain or projecting the 360-degree depth output from the 360 model to patches and computing in the perspective domain. Since training is conducted in an up-to-scale relative depth manner, stitching the patch of perspective images back to equirectangular with an aligned scale will lead to failure in training Figure 4, which is an additional research topic that is worth investigation. We opt to compute the loss in the perspective domain, facilitating faster and easier training without the need for additional alignment optimization.

### 3.2 Random Rotation Processing

Directly applying Depth Anything on cube-projected unlabeled data does not yield improvements due to ignorance of cross-cube-face relation, leading to cube artifacts (Figure 5). This issue arises from the separate estimation of perspective cube faces, where monocular depth is estimated based on semantic information, lacking a comprehensive understanding of the entire scene. To address this, we propose a random rotation preprocessing step in front of the perspective foundation model.

As depicted in Figure 2, the rotation is applied to equirectangular projection RGB images using a random rotation matrix, followed by cube projection. This results in a more diverse set of cube faces, capturing relative distances between ceilings, walls, windows, and other objects more effectively. With the proposed random rotation technique, knowledge distillation becomes more comprehensive as the point of view is not static. The inference by the perspective foundation model is performed on the fly, with parameters frozen during the training of the 360 model.

In order to perform random rotation, we apply a rotation matrix on the equirectangular coordinates, noted as $(\theta, \phi)$, and rotation matrix as $\mathcal{R}$.

$$(\hat{\theta}, \hat{\phi}) = \mathcal{R} \cdot (\theta, \phi). \tag{1}$$

For equirectangular to cube projection, the field-of-view (FoV) of each cube face is equal to 90 degrees; each face can be considered as a perspective camera whose focal length is $w/2$, and all faces share the same center point in the world coordinate. Since the six cube faces share the same center point, the extrinsic matrix of each camera can be defined by a rotation matrix $R_i$. $p$ is then the pixel

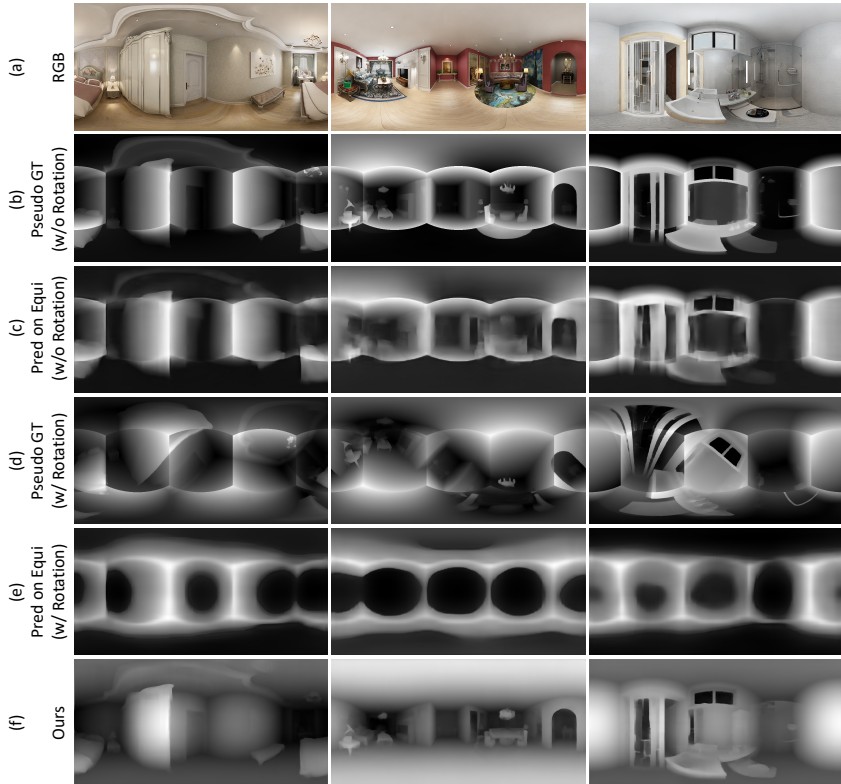

Figure 4: **Qualitative visualization of a model trained directly on pseudo equirectangular data without scale alignment**. We propose calculating the loss with pseudo ground truth on cube faces due to scale misalignment between the six faces during the cube-to-equirectangular projection. We showcase the results of a model trained on pseudo equirectangular data without scale alignment as a simple baseline to demonstrate the importance of calculating loss separately on each of the six faces. The images are presented from top to bottom as follows: (a) RGB images. (b) Pseudo cube ground truth projected directly to equirectangular. (c) Prediction trained with row 2. (d) Pseudo cube ground truth with rotation projected directly to equirectangular. (e) Prediction trained with row 4. (f) Our model's predictions are trained on cube faces separately with rotation.

on the cube face

$$p = K \cdot R_i^T \cdot q, \tag{2}$$

where,

$$q = \begin{bmatrix} q_x \\ q_y \\ q_z \end{bmatrix} = \begin{bmatrix} sin(\theta) \cdot \cos(\phi) \\ \sin(\phi) \\ \cos\theta \cdot \cos\phi \end{bmatrix}, K = \begin{bmatrix} w/2 & 0 & w/2 \\ 0 & w/2 & w/2 \\ 0 & 0 & 1 \end{bmatrix}, \tag{3}$$

where $\theta$ and $\phi$ are longitude and latitude in equirectangular projection and q is the position in Euclidean space coordinates.

### 3.3 Loss Function

The training process closely resembles that of MiDaS, Depth Anything, and other cross-dataset methods. Our goal is to provide depth estimation for any 360-degree images. Following previous approaches that trained on multiple datasets, our training objective is to estimate relative depth. The depth values are first transformed into disparity space using the formula $1/d$ and then normalized to the range $[0, 1]$ for each disparity map.

To adapt to cross-dataset training and pseudo ground truths from the foundation model, we employed the affine-invariant loss, consistent with prior cross-dataset methodologies. This loss function disregards absolute scale and shifts for each domain, allowing for effective adaptation across different

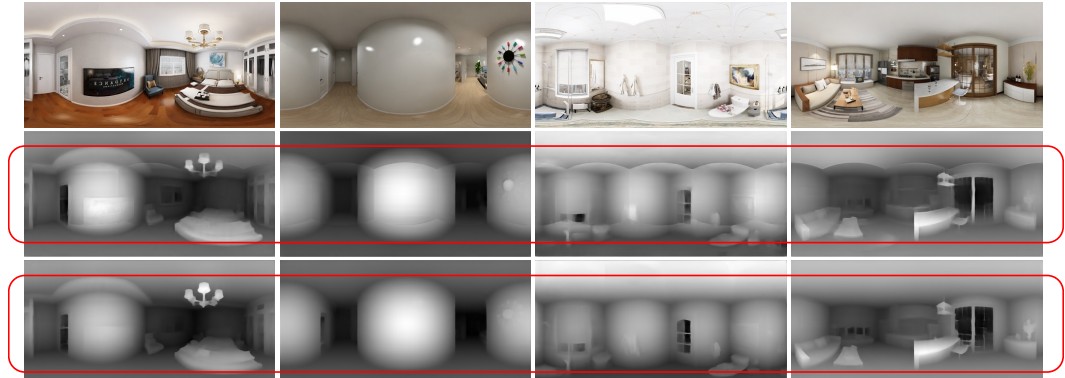

Figure 5: **Cube Artifact.** Shown in the center row of the figure, an undesired cube artifact appears when we apply joint training with pseudo ground truth from Depth Anything [59] directly. This issue arises from independent relative distances within each cube face caused by a static point of view. Ignoring cross-cube relationships results in poor knowledge distillation. To address this, as shown in Figure 2(c), we randomly rotate the RGB image before inputting it into Depth Anything. This enables better distillation of depth information from varying perspectives within the equirectangular image.

datasets and models.

$$\mathcal{L}1 = \frac{1}{HW} \sum_{i=1}^{HW} \rho(d_i^*, d_i), \tag{4}$$

where $d_i^*$ and $d_i$ are the prediction and ground truth, respectively. $\rho$ represents the affine-invariant mean absolute error loss:

$$\rho(d_i^*, d_i) = |\hat{d}_i^* - \hat{d}_i|. \tag{5}$$

Here, $\hat{d}_i$ and $\hat{d}_i^*$ are the scaled and shifted versions of the prediction $d_i^*$ and ground truth $d_i$:

$$\hat{d}_i = \frac{d_i - t(d)}{s(d)}, \tag{6}$$

where $t(d)$ and $s(d)$ are used to align the prediction and ground truth to have zero translation and unit scale:

$$t(d) = \text{median}(d), \quad s(d) = \frac{1}{HW} \sum_{i=1}^{HW} |d_i - t(d)|. \tag{7}$$

## 4 Experiments

These notations apply for all tables: **M**: Matterport3D [6], **SF**: Stanford2D3D [2], **ST**: Structured3D [65], **SP**: Spatialaudiogen [29], **-all** indicates using the entire train, validation, and test sets of the specific dataset, and **(p)** denotes using pseudo ground truth generated by Depth Anything [59]. Due to space limits, we provide the experimental setup in the appendix, including implementation details and evaluation metrics.

### 4.1 Baselines

Recent state-of-the-art methods [1, 64, 47, 48, 16, 31, 40] have emerged. We chose UniFuse and BiFuse++ as our baseline models for experiments, as many of the aforementioned methods did not fully release pre-trained models or provide training code and implementation details. It's worth noting that PanoFormer [40] is not included due to incorrect evaluation code and results. Both selected models are re-implemented with affine-invariant loss on disparity for a fair comparison and to demonstrate improvement. We conduct experiments on the Matterport3D [6] benchmark to demonstrate performance gains within the same dataset/domain, and we perform zero-shot evaluation on the Stanford2D3D [2] test set to demonstrate the generalization capability of our proposed training technique. To further validate its robustness, we evaluate additional baseline models [45, 64] in zero-shot setting, showcasing the effectiveness of our approach for non-dual-projection models.

Table 2: **Matterport3D Benchmark.** The upper section lists 360 methods trained with metric depths in meters using BerHu loss. All numbers are sourced from their respective papers. The lower section includes selected methods retrained with relative depth (disparity) using affine-invariant loss.

| Method | Loss | Train | Test | Abs Rel $\downarrow$ | $\delta_1 \uparrow$ | $\delta_2 \uparrow$ | $\delta_3 \uparrow$ |
|--------|------|-------|------|---------|----------|----------|----------|
| BiFuse [47] | BerHu | M | M | - | 0.845 | 0.932 | 0.963 |
| UniFuse [16] | BerHu | M | M | 0.106 | 0.890 | 0.962 | 0.983 |
| SliceNet [31] | BerHu | M | M | - | 0.872 | 0.948 | 0.972 |
| BiFuse++ [48] | BerHu | M | M | - | 0.879 | 0.952 | 0.977 |
| HRDFuse [1] | BerHu | M | M | 0.097 | 0.916 | 0.967 | 0.984 |
| UniFuse [16] | Affine-Inv | M | M | 0.102 | 0.893 | 0.970 | 0.989 |
| UniFuse [16] | Affine-Inv | M, ST-all (p) | M | 0.089 | 0.911 | 0.975 | 0.991 |
| BiFuse++ [48] | Affine-Inv | M | M | 0.094 | 0.914 | 0.974 | 0.989 |
| BiFuse++ [48] | Affine-Inv | M, ST-all (p) | M | **0.085** | **0.917** | **0.976** | **0.991** |

## 4.2 Benchmarks Evaluation

We conducted our in-domain improvement experiment on the widely used 360-degree depth benchmark, Matterport3D [6], to showcase the results of perspective foundation model distillation on the two selected baseline models, UniFuse [16] and BiFuse++[48]. In Table 2, we list the metric depth evaluation results from state-of-the-art methods on this benchmark. Subsequently, we present the re-trained baseline models using affine-invariant loss on disparity to ensure a fair comparison with their original depth metric training. Finally, we demonstrate the improvement achieved with results trained on the labeled Matterport3D training set and the entire Structured3D dataset with pseudo ground truth.

## 4.3 Zero-Shot Evaluation

Our goal is to estimate depths for all 360-degree images, making zero-shot performance crucial. Following previous works [47, 16], we adopted their zero-shot comparison setting, where models trained on the entire Matterport3D [6] dataset are tested on the Stanford2D3D [2] test set. In Table 3, the upper section lists methods trained with metric depth ground truth, with numbers sourced from their respective papers. The lower section includes models trained with affine-invariant loss on disparity ground truth. As shown in Figure 6, [16, 48] demonstrate generalization improvements with a lower error on the Stanford2D3D dataset.

Depth Anything [59] and Marigold [19] are state-of-the-art zero-shot depth models trained with perspective depths. As shown in Table 3, due to the domain gap and different camera projections, foundation models trained with perspective depth cannot be directly applied to 360-degree images. We demonstrated the zero-shot improvement on UniFuse [16], BiFuse++ [48] and non-dual-projection methods [45, 64] with models trained on the entire Matterport3D [6] dataset with ground truth and the entire Structured3D [65] or SpatialAudioGen [29] dataset with pseudo ground truth generated using Depth Anything [59].

As Structured3D provides ground truth labels for its dataset, we also evaluate our models on its test set to assess how well they perform with pseudo labels. Table 4 shows the improvements achieved on the Structured3D test set when using models trained with pseudo labels. It's worth noting that even when the 360 model is trained on pseudo labels from SpatialAudioGen, it performs similarly well. This demonstrates the success of our distillation technique and the model's ability to generalize across different datasets.

## 4.4 Qualtative Results in the Wild

We demonstrated the qualitative results in Figure 8 and Figure 7 360-degree images that were either captured by us or downloaded from the internet[1]. These examples showcase the zero-shot capability of our model when applied to data outside the aforementioned 360-degree datasets.

---

[1]Stig Nygaard, https://www.flickr.com/photos/stignygaard/49659694937, CC BY 2.0 DEED
Dominic Alves, https://www.flickr.com/photos/dominicspics/28296671029/, CC BY 2.0 DEED
Luca Biada, https://www.flickr.com/photos/pedroscreamerovsky/6873256488/, CC BY 2.0 DEED
Luca Biada, https://www.flickr.com/photos/pedroscreamerovsky/6798474782/, CC BY 2.0 DEED

Table 3: **Zero-shot Evaluation on Stanford2D3D.** We perform zero-shot evaluations with models trained on other datasets. Following the original training settings, we train the 360 models [48, 16, 45, 64] on the entire Matterport3D dataset and then test on Stanford3D's test set.

| Method | Loss | train | test | Abs Rel ↓ | $\delta_1 \uparrow$ | $\delta_2 \uparrow$ | $\delta_3 \uparrow$ |
|---|---|---|---|---|---|---|---|
| BiFuse [47] | BerHu | M-all | SF | 0.120 | 0.862 | - | - |
| UniFuse [16] | BerHu | M-all | SF | 0.094 | 0.913 | - | - |
| BiFuse++ [48] | BerHu | M-all | SF | 0.107 | 0.914 | 0.975 | 0.989 |
| Depth Anything [59] | Affine-Inv | Pers. | SF | 0.248 | 0.635 | 0.899 | 0.97 |
| Marigold [19] | Affine-Inv | Pers. | SF | 0.195 | 0.692 | 0.942 | 0.982 |
| UniFuse [16] | Affine-Inv | M-all | SF | 0.090 | 0.914 | 0.976 | 0.990 |
| UniFuse [16] | Affine-Inv | M-all, ST-all (p) | SF | 0.086 | 0.924 | 0.977 | 0.990 |
| UniFuse [16] | Affine-Inv | M-all, SP-all (p) | SF | 0.090 | 0.920 | 0.978 | 0.990 |
| BiFuse++ [48] | Affine-Inv | M-all | SF | 0.090 | 0.921 | 0.976 | 0.990 |
| BiFuse++ [48] | Affine-Inv | M-all, ST-all (p) | SF | **0.082** | **0.931** | **0.979** | 0.991 |
| BiFuse++ [48] | Affine-Inv | M-all, SP-all (p) | SF | 0.086 | 0.926 | **0.979** | 0.991 |
| HoHoNet [45] | Affine-Inv | M-all | SF | 0.095 | 0.906 | 0.975 | 0.991 |
| HoHoNet [45] | Affine-Inv | M-all, ST-all (p) | SF | 0.088 | 0.920 | **0.979** | **0.992** |
| EGFormer [64] | Affine-Inv | M-all | SF | 0.098 | 0.906 | 0.972 | 0.989 |
| EGFormer [64] | Affine-Inv | M-all, ST-all (p) | SF | 0.086 | 0.923 | 0.976 | 0.990 |

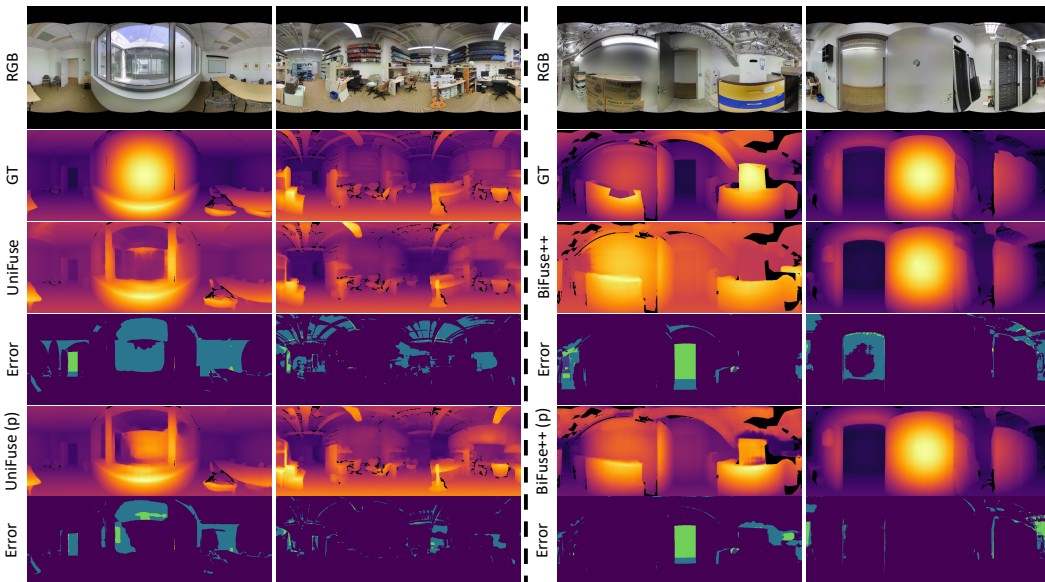

Figure 6: **Zero-shot qualitative with UniFuse [16] (*left*) and BiFuse++ [48] (*right*) tested on Stanford2D3D.**

Table 4: **Structured3D Test Set.** We demonstrate the improvement on the Structured3D test set using pseudo ground truth for training. The lower section shows enhancements with models trained on pseudo ground truth from Matterport3D and SpatialAudioGen, indicating similar improvements. This highlights the successful distillation of Depth Anything.

| Method | Loss | train | test | Abs Rel ↓ | $\delta_1 \uparrow$ | $\delta_2 \uparrow$ | $\delta_3 \uparrow$ |
|---|---|---|---|---|---|---|---|
| UniFuse [16] | Affine-Inv | M-all | ST | 0.202 | 0.759 | 0.932 | 0.970 |
| UniFuse [16] | Affine-Inv | M-all, ST-all (p) | ST | 0.130 | 0.887 | 0.953 | 0.977 |
| UniFuse [16] | Affine-Inv | M-all, SP-all (p) | ST | 0.152 | 0.864 | 0.946 | 0.972 |

Table 5: **Metric depth fine-tuning.** We fine-tune our model trained with Matterport3D [6] ground truth label and Structured3D [65] pseudo label on relative depth with Stanford2D3D [2]'s training set metric depths with a single epoch.

| Method | MAE ↓ | Abs Rel ↓ | RMSE ↓ | RMSElog ↓ | $\delta_1$ ↑ | $\delta_2$ ↑ | $\delta_3$ ↑ |
|---|---|---|---|---|---|---|---|
| UniFuse [16] | 0.208 | 0.111 | 0.369 | 0.072 | 0.871 | 0.966 | 0.988 |
| UniFuse [16] (Ours) | 0.206 | 0.118 | 0.351 | 0.049 | 0.910 | 0.971 | 0.987 |

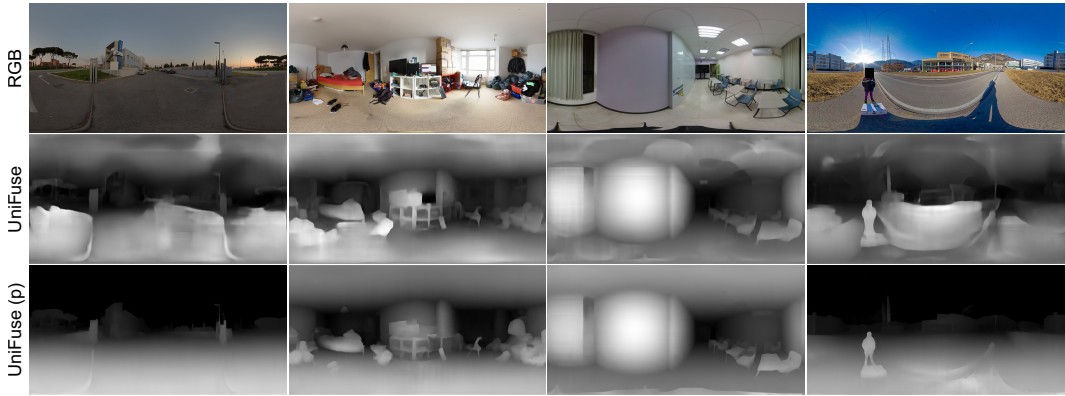

Figure 7: **Generalization ability in the wild with depth map visualization.** We showcase zero-shot qualitative results using a combination of images we captured and randomly sourced from the internet to assess the model's generalization ability. For privacy reasons, we have obscured the cameraman in the images.

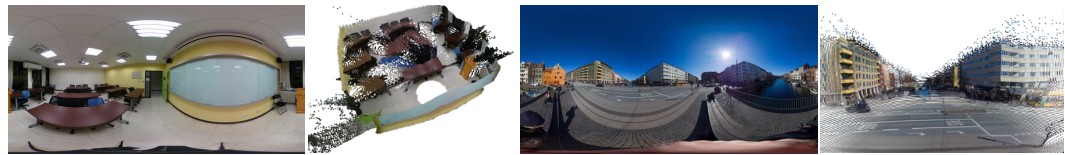

Figure 8: **Generalization ability in the wild with point cloud visualization.** We showcase zero-shot qualitative results in the point cloud using a combination of images we captured and randomly sourced from the internet to assess the model's generalization ability.

## 4.5 Fine-Tuned to Metric Depth Estimation

We use our pre-trained model as an initial weight and fine-tune on Stanford2D3D [2] metric depth to demonstrate the effectiveness of our pre-trained relative depth model's ability to adapt to metric depth with a single epoch in Table 5

## 5 Conclusion

Our proposed method significantly advances 360-degree monocular depth estimation by leveraging perspective models for pseudo-label generation on unlabeled data. The use of cube projection with random rotation and affine-invariant loss ensures robust training and improved depth prediction accuracy while bridging the domain gap between perspective and equirectangular projection. By effectively addressing the challenges of limited labeled data with cross-domain distillation, our approach opens new possibilities for accurate depth estimation in 360 imagery. This work lays the groundwork for future research and applications, offering a promising direction for further advancements in 360-degree depth estimation.

**Limitations.** Our work faces limitations due to its heavy reliance on the quality of unlabeled data and pseudo labels from perspective foundation models. The results are significantly impacted by data quality (Section 3.1). Without data cleaning, the training process resulted in NaN values. Another limitation is that although with unlabeled data, the scarcity of data still exists compared to other tasks.

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

# A Appendix / Supplemental Material

## A.1 Experimental Setup

**Implementation details.** Our work is divided into two stages: (1) offline mask generation and (2) online joint training. (1) In the first stage, we use Grounded-Segment-Anything [36], which combines state-of-the-art detection and segmentation models. We set the `BOX_THRESHOLD` and `TEXT_THRESHOLD` to 0.3 and 0.25, respectively, following the recommendations of the official code. We use "sky" and "watermark" as text prompts. All pixels with these labels are set to False to form our valid mask for the second stage of training. (2) In the second stage, each batch consists of an equal mix of labeled and unlabeled data. We follow the backbone model's official settings for batch size, learning rate, optimizer, augmentation, and other hyperparameters, changing only the loss function to affine-invariant loss. Unlike Depth Anything, which sets invalid sky regions to zero disparity, we ignore these invalid pixels during loss calculation, consistent with ground truth training settings. We average the loss for ground truth and pseudo ground truth during updates. All our experiments are conducted on a single RTX 4090, both offline and online. However, if future 360-degree state-of-the-art methods or perspective foundation models require higher VRAM usage, the computational resource requirements may increase.

**Metrics.** In line with previous cross-dataset works, all evaluation metrics are presented in percentage terms. The primary metric is Absolute Mean Relative Error (AbsRel), calculated as: $\frac{1}{M}\sum_{i=1}^{M}|a_i - d_i|/d_i$, where $M$ is the total number of pixels, $a_i$ is the predicted depth, and $d_i$ is the ground truth depth. The second metric, $\delta_j$ accuracy, measures the proportion of pixels where $max(ai/di, di/ai)$ is within $1.25^j$. During evaluations, we follow [16, 47, 48] to ignore areas where ground truth depth values are larger than 10 or equal to 0. Given the ambiguous scale of self-training results, we apply median alignment after converting disparity output to depth before evaluation, as per the method used in [67]:

$$d' = d \cdot \frac{\mathrm{median}(\hat{d})}{\mathrm{median}(d)}, \tag{8}$$

where $d$ is the predicted depth from inverse disparity and $\hat{d}$ is the ground truth depth. This ensures a fair comparison by aligning the median depth values of predictions and ground truths.

## A.2 More Qualitative

We demonstrate additional zero-shot qualitative results in Figure 9 and in-the-wild results in Figure 12. In-domain results on the Matterport3D test sets are showcased in Figure 10 and Figure 11.

## A.3 Dataset Statistic

As described in Sec.3.1 of the main paper, there is a significant difference in the number of images between the perspective and equirectangular datasets. Detailed statistics of the datasets are listed in Table 6.

## A.4 Ground Truth and Pseudo Label Ratio Ablation

Unlike many previous knowledge distillation approaches that use a higher proportion of pseudo labels during model training, we opt for an equal ratio of ground truth to pseudo labels. Through an ablation study exploring the relationship between this data ratio and model performance, we observe a robust improvement starting from 1 : 1 in Table 7.

## A.5 Perspective Camera Projection Ablation

There are various perspective camera projections for panoramic imagery, with cube and tangent image projections being the most common, both widely used in previous works. In Table 8, we compare these two projections and observe similar improvements when applying our proposed training pipeline, demonstrating the effectiveness of our method. For robust knowledge distillation in relative depth estimation, we select the cube projection due to its wider field of view coverage.

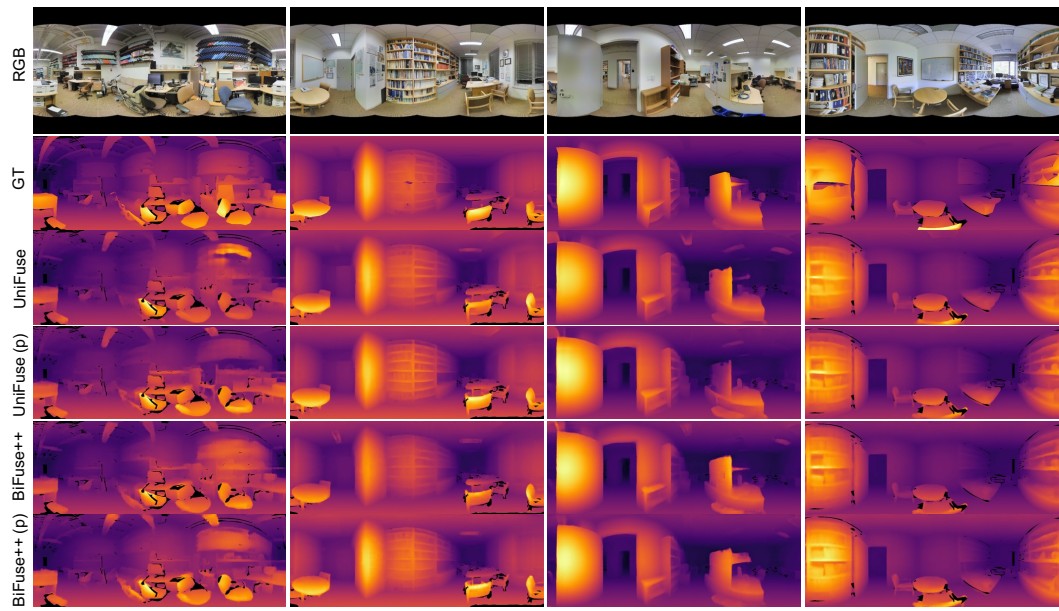

Figure 9: **More qualitative tested on Stanford2D3D with zero-shot setting.**

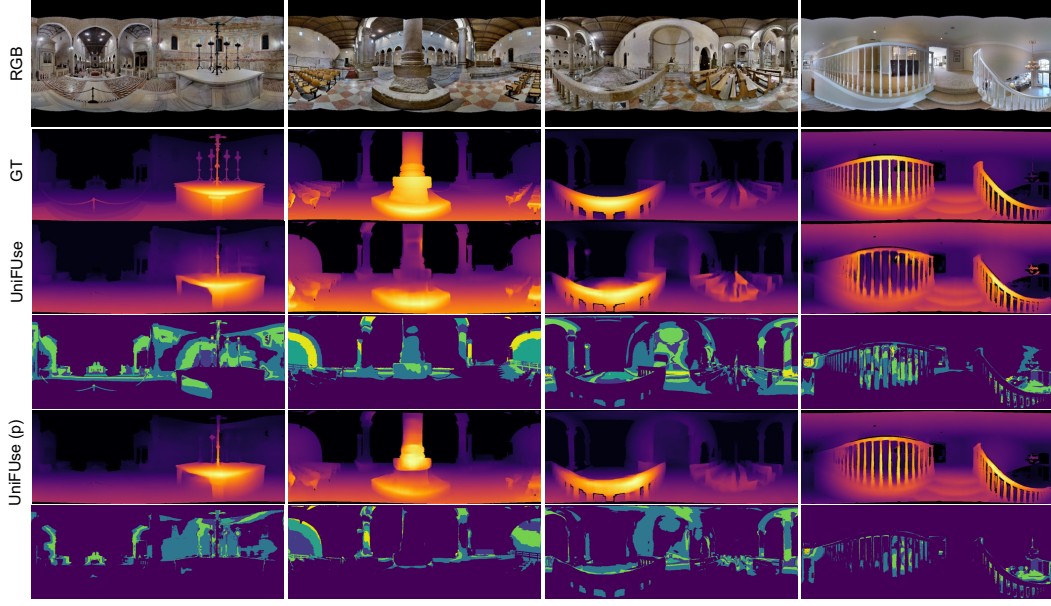

Figure 10: **In-domain qualitative with UniFuse.**

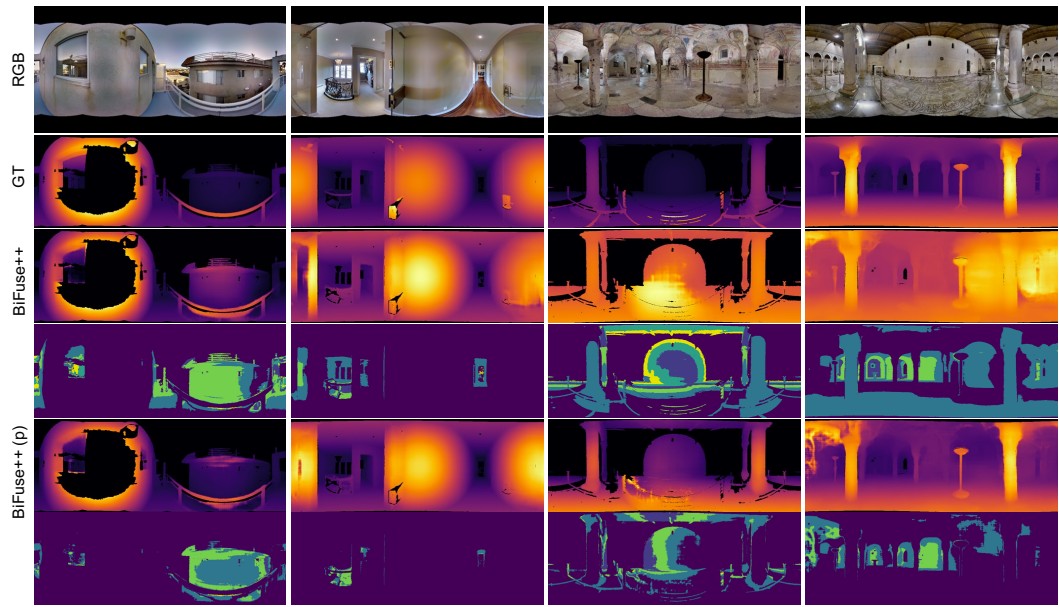

Figure 11: **In-domain qualitative with BiFuse++.**

Table 6: **360 monocular depth estimation lacks a large amount of training data.** This table lists datasets used in 360-degree monocular depth estimation alongside perspective depth datasets from the Depth Anything methodology. The volume of training data for 360-degree imagery (*right column*) is significantly smaller than that for perspective imagery (*left column*) by about 200 times. This highlights the need for using perspective distillation techniques to enhance the limited data available for 360-degree depth estimation. Ground truth (GT) labels are noted where applicable, showing the available resources for training in these domains.

| Dataset | Perspective Venue | # of images | GT labels | Dataset | Equirectangular Venue | # of images | GT labels |
|---|---|---|---|---|---|---|---|
| MegaDepth [24] | CVPR 2018 | 128K | ✓ | Stanford2D3D [2] | arXiv 2017 | 1.4K | ✓ |
| TartanAir [52] | IROS 2020 | 306K | ✓ | Matterport3D [6] | 3DV 2017 | 10.8K | ✓ |
| DIML [10] | arXiv 2021 | 927K | ✓ | Structured3D [65] | ECCV 2020 | 21.8K | ✓ |
| BlendedMVS [60] | CVRP 2020 | 115 K | ✓ | SpatialAudioGen [29] | NeurIPS 2018 | 344K | - |
| HRWSI [55] | CVPR 2020 | 20K | ✓ | | | | |
| IRS [51] | ICME 2021 | 103K | ✓ | | | | |
| ImageNet-21K [38] | IJCV 2015 | 13.1M | - | | | | |
| BDD100K [63] | CVPR 2020 | 8.2M | - | | | | |
| Google Landmarks [53] | CVPR 2020 | 4.1M | - | | | | |
| LSUN [62] | arXiv 2015 | 9.8M | - | | | | |
| Objects365 [39] | ICCV 2019 | 1.7M | - | | | | |
| Open Images V7 [21] | IJCV 2020 | 7.8M | - | | | | |
| Places365 [66] | TPAMI 2017 | 6.5M | - | | | | |
| SA-1B [20] | ICCV 2023 | 11.1M | - | | | | |

Table 7: **Ratios of GT and Pseudo label during training.** We conduct additional experiments with varying ratios, which shows our method is robust across different ratios starting from 1:1.

| Ratio | train(GT) | train(Pseudo) | test | Abs Rel ↓ | $\delta_1$ ↑ | $\delta_2$ ↑ | $\delta_3$ ↑ |
|---|---|---|---|---|---|---|---|
| 1:1 | M-all | ST-all(p) | SF | 0.086 | 0.924 | 0.977 | 0.990 |
| 1:2 | M-all | ST-all(p) | SF | 0.087 | 0.923 | 0.977 | 0.990 |
| 1:4 | M-all | ST-all(p) | SF | 0.085 | 0.923 | 0.977 | 0.990 |

Table 8: **Comparison between Tangent Image and Cube Projection.** We compare two of the most commonly used perspective camera projections for panorama images. As shown in the table, both projections yield similar quantitative results. However, we select the cube projection for knowledge distillation due to its broader field of view coverage.

| Projection | Method | train | test | Abs Rel ↓ | $\delta_1 \uparrow$ | $\delta_2 \uparrow$ | $\delta_3 \uparrow$ |
|---|---|---|---|---|---|---|---|
| Cube | UniFuse | M-all+ST-all(p) | SF | **0.086** | **0.924** | 0.977 | 0.990 |
| Tangent | UniFuse | M-all+ST-all(p) | SF | 0.087 | 0.923 | **0.978** | **0.991** |

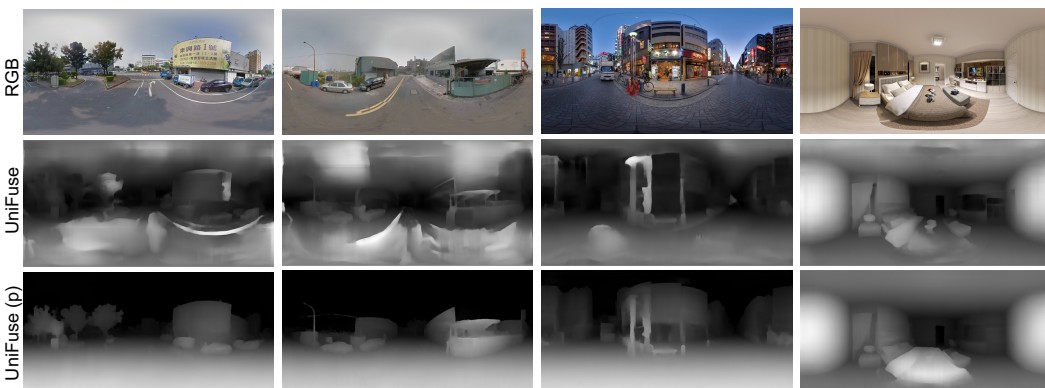

Figure 12: **Additional in-the-wild results.** We compare our proposed joint-training method (Matterport3D (GT) + SpatialAudioGen (Pseudo)) with a model trained only on the Matterport3D dataset, using data randomly downloaded from the internet. This comparison demonstrates the significant improvement of our method along with its generalization ability and effectiveness on real-world data.

