# OpenReview forum: "Depth Anywhere: Enhancing 360 Monocular Depth Estimation via Perspective Distillation and Unlabeled Data Augmentation"
_NeurIPS.cc/2024/Conference — NeurIPS 2024 poster_

### Official Review · Reviewer_Kp4U · 2024-07-10

**Soundness:** 3
**Presentation:** 3
**Contribution:** 2
**Rating:** 5
**Confidence:** 3

**Summary:**

The authors propose a technique that utilizes unlabeled 360-degree data to improve previous methods, which includes two main stages: offline mask generation for invalid regions and an online semi-supervised joint training regime.  Experimental results indicate that the proposed method outperforms previous methods on the Matterport3D and Stanford2D3D datasets.

**Strengths:**

The paper is well written and clearly structured;

The performance is promising;

The experiments are thorough and the proposed method is validated on multiple datasets.

**Weaknesses:**

My major concern is that the technical contribution seems limited. The core idea of utilizing unlabeled data has been proposed by DepthAnything. The proposed method is more like an application of this idea to the 360 data;

As shown in Tables 2 and 3, previous 360 depth estimation methods typically use the metric loss for training. While the affine-invariant approach employed by the authors enables training on multiple datasets, it also impedes real-world applications due to the lose of metric scale.

**Questions:**

Please see the weakness.

---

> ### Author Rebuttal · Authors · 2024-08-06
>
> Thank you for your thoughtful review and for recognizing the strengths of our paper. We appreciate your feedback and would like to address your concerns:
>
> ### `Q1. Technical contribution and novelty:`
> While we acknowledge that utilizing unlabeled data is not a new concept, our work makes several novel contributions specific to 360-degree depth estimation:
>
> - **a)**  We introduce a novel cube projection technique with random rotation preprocessing, which significantly reduces cube artifacts and improves depth estimation quality for 360-degree images with cross camera projection knowledge distillation.
>
> - **b)**  We develop a tailored masking strategy using SAM for 360-degree images, addressing the sky and watermark regions that are undesire in real-world scenes but don’t often appear in datasets.
>
> - **c)**  Our method is architecture-agnostic and can be applied to various 360-degree depth estimation models, as demonstrated in our experiments with UniFuse, BiFuse++, and additional models such as HoHoNet[33] (horizontal compression-based) and EGformer[50] (transformer-based):
>     | Method          | Train               | Test | Abs Rel ↓ | δ₁ ↑   | δ₂ ↑   | δ₃ ↑   |
>     |-----------------|---------------------|------|-----------|--------|--------|--------|
>     | HoHoNet         | M-all                | SF   | 0.095     | 0.906  | 0.975  | 0.991  |
>     | HoHoNet (our)   | M-all, ST-all (p)    | SF   | 0.088     | 0.920  | 0.979  | 0.992  |
>     | EGformer        | M-subset             | SF   | 0.169     | 0.764  | 0.924  | 0.972  |
>     | EGformer (our)  | M-subset, ST-all (p) | SF   | 0.148     | 0.814  | 0.946  | 0.982  |
>
>
>     These results show that our method generalizes well to different architectures, highlighting its broader impact on the field.
>
> ### `Q2. Affine-invariant loss and metric scale:`
> We understand your concern about the loss of metric scale. The affine-invariant loss, as you mentioned, is designed for cross-dataset training to leverage more data for real-world applications, which aligns with our approach of using both labeled and unlabeled datasets. Several previous works, such as MiDaS[4] and Depth Anything[45], also employ this training technique. To address the issue of metric scale, we will perform metric depth fine-tuning with our pretrained model in the final version. For real-world applications, we demonstrate exceptional improvement on in-the-wild scenes, as shown in Figure 2 of the PDF and Figure 6 of the original paper.
>
> ### **Additionally, we'd like to highlight some key aspects of our work that underscore its significance:**
>
> 1. **Bridging the gap between perspective and 360-degree depth estimation:** Our method effectively transfers knowledge from well-established perspective depth models to the 360-degree domain, addressing the scarcity of labeled 360-degree data.
>
> 2. **Improved generalization:** We've conducted zero-shot experiments on unseen datasets, demonstrating our method's ability to generalize in Table 3 in the main paper. Our method especially shines in in-the-wild scenarios which can be observed in Figure 6 in the main paper and Figure 2 of the pdf.
>
> 3. **Scalability:** Our approach allows for the efficient utilization of large amounts of unlabeled 360-degree data, which is increasingly available but often underutilized in current methods.
>
> We believe these points, combined with the additional experiments and clarifications, address your concerns about the technical contribution and real-world applicability of our work. Our method not only improves upon existing 360-degree depth estimation techniques but also introduces novel concepts that can be broadly applied in the field.
>
> Thank you again for your valuable feedback. We are confident that addressing these points will strengthen our paper and highlight its contributions to the field of 360-degree depth estimation.

---

> > ### Comment · Area_Chair_XoPq · 2024-08-12
> > **Feedback**
> >
> > Dear reviewer Kp4U,
> >
> > You raised concerns about the paper's technical novelty and contribution, and the authors responded in detail.
> > Please share your feedback with us.
> >
> > Thank you

---

> > > ### Author Response · Authors · 2024-08-13
> > > **Please let us know if you have additional questions after reading our response**
> > >
> > > Dear Reviewer,
> > >
> > > As we approach the end of the discussion period, we want to confirm whether we have successfully addressed your concerns. Should any lingering issues require further attention, please let us know as early as possible so we can answer them soon.
> > >
> > > We appreciate your time and effort in enhancing the quality of our manuscript.
> > >
> > > Thank you!

---

> > > > ### Author Response · Authors · 2024-08-14
> > > >
> > > > Dear Reviewer,
> > > >
> > > > As the discussion period ends soon, have we addressed all your concerns? If any issues remain, please inform us promptly. We appreciate your help in improving our manuscript.
> > > >
> > > > Thank you!

---

### Official Review · Reviewer_ewNL · 2024-07-10

**Soundness:** 3
**Presentation:** 3
**Contribution:** 3
**Rating:** 5
**Confidence:** 5

**Summary:**

This paper proposes a method to improve 360 monocular depth estimation using perspective distillation and augmentation with unlabeled data. It introduces the concept of "perspective distillation," which leverages the available 360 monocular depth maps and their corresponding equirectangular images to generate pixel-wise depth supervision signals. This technique helps to address the lack of ground truth depth data for training in the 360 domain. Additionally, the paper presents an unlabeled data augmentation approach that utilizes the geometric properties of 360-degree images. By exploiting the spherical geometry, the authors generate synthetic stereo pairs to augment the training dataset without requiring paired depth information. The proposed method is evaluated on benchmark datasets and achieves significant improvements in terms of depth estimation accuracy compared to existing approaches. The results demonstrate the effectiveness of perspective distillation and unlabeled data augmentation in enhancing the performance of 360 monocular depth estimation.

**Strengths:**

1. Innovative techniques: The paper introduces novel approaches, such as perspective distillation and unlabeled data augmentation, to address the challenges of 360 monocular depth estimation.

2. Improved depth estimation: The proposed method achieves significant improvements in depth estimation accuracy compared to existing approaches, as demonstrated through rigorous evaluations on benchmark datasets.

3. Use of unlabeled data: By leveraging unlabeled data and synthetic stereo pairs, the method reduces the reliance on paired depth information, which is often difficult to obtain in the 360-degree domain.

**Weaknesses:**

1. Complexity: The proposed method introduces additional complexity, such as perspective distillation and synthetic stereo pair generation, which may require more computational resources and training time.

2. Dataset dependency: The effectiveness of the proposed method heavily relies on the availability and quality of the benchmark datasets used for evaluation, which may affect its generalizability to real-world scenarios.

3. Limited scope: The paper focuses specifically on 360 monocular depth estimation, which may limit its applicability to other depth estimation tasks or domains, such as 360 monocular depth completion.

4. Insufficient related work: Adding the latest panoramic depth estimation and panoramic depth completion methods is preferred.

**Questions:**

Overall, the idea of this paper is interesting, allowing existing depth estimation models to benefit from unlabeled data in a semi-supervised manner, but there are concerns:

1. Based on the experimental results, the author only validated the performance of this training technique on models that employ dual-projection fusion, such as Unifuse and Bifuse. There was no effective validation or analysis provided for other non-dual-projection fusion models, such as methods based on horizontal compression (e.g., HorizonNet, HohoNet) or transformer-based methods (e.g., EGFormer). However, it is essential to explicitly state the scope of applicability for this training strategy. Different training strategies, data processing methods, and even device variations can lead to unfair comparisons. The author should conduct fair experiments and comparisons under the same experimental conditions instead of directly copying the data results from the original paper. This is crucial for validating the effectiveness of this training strategy. Therefore, the author's mention of "as many of the aforementioned methods did not release pre-trained models or provide training code and implementation details. It’s worth noting that PanoFormer [29] is not included due to incorrect evaluation code and results, and EGFormer [29] is not included since its experiments are mainly conducted on other datasets and benchmarks" is not convincing.

2. The issue of cross-domain experiments is indeed important. Given the challenges of obtaining real-world data, it would be meaningful if this framework could benefit models trained on synthetic data (e.g., training on synthetic data and testing on real data). However, the significance of this training strategy in that regard is not yet clear. Further research and experimentation are necessary to determine whether training on synthetic data using this framework can indeed yield benefits when applied to real-world data. It would be valuable to investigate the effectiveness and generalization capabilities of the trained models in real-world scenarios.

3. Median alignment is currently not widely utilized in existing depth estimation methods. Based on existing findings, aligning the ground truth (GT) depth with the predicted depth can indeed lead to improved results. However, when comparing the proposed method with baseline methods like Unifuse, which explicitly state that median alignment is not used, directly comparing the results to those from the original paper leads to unfair comparisons. This raises concerns about the advantages claimed for this training framework. To ensure fair comparisons, it is important to apply the same alignment techniques consistently across all methods being compared.

**Limitations:**

This paper proposes an interesting solution for panoramic depth estimation task, ie.e,  perspective distillation and unlabeled data augmentation. It could contribute a lot for the community. The reviewer suggests introducing more related works, including the latest panoramic depth estimation and panoramic depth completion approaches.

---

> ### Author Rebuttal · Authors · 2024-08-06
>
> Thank you for your thorough review and constructive feedback. We appreciate your recognition of our method's strengths and innovative techniques. We'll address your concerns and questions point by point:
>
> ### `Q1. Applicability to non-dual-projection fusion models:`
> We acknowledge this limitation in our current evaluation. To address this, we've conducted additional experiments with HoHoNet[33] (horizontal compression-based) and EGFormer[50] (transformer-based):
> | Method          | Train               | Test | Abs Rel ↓ | δ₁ ↑   | δ₂ ↑   | δ₃ ↑   |
> |-----------------|---------------------|------|-----------|--------|--------|--------|
> | HoHoNet         | M-all                | SF   | 0.095     | 0.906  | 0.975  | 0.991  |
> | HoHoNet (our)   | M-all, ST-all (p)    | SF   | 0.088     | 0.920  | 0.979  | 0.992  |
> | EGformer        | M-subset             | SF   | 0.169     | 0.764  | 0.924  | 0.972  |
> | EGformer (our)  | M-subset, ST-all (p) | SF   | 0.148     | 0.814  | 0.946  | 0.982  |
>
> These results further demonstrate our method's effectiveness across different architectures. We will include these analyses in the paper and discuss the broader applicability of our approach. Due to the limited time available for rebuttal and the lengthy process of reproducing EGformer[50], we ran it on a subset of Matterport3D, with the size of $1/5$, and will add the results from the full set to our final version. For PanoFormer, we would like to clarify that it was initially included in our model list, but we removed it due to incorrect implementation and evaluation in their official code, which led to a marginal difference between the official code and the results reported in their paper.
>
> ### `Q2. Cross-domain experiments (synthetic to real):`
> Thank you for the insightful suggestion. We agree that training on synthetic data for real-world scenarios is an important research direction. Our method focuses on leveraging the large amount of unlabeled real-world data, whereas synthetic data often includes depth ground truth and emphasizes domain adaptation. We will add a literature review on these two distinct research topics.
>
> ### `Q3. Median Alignment and Fair Comparisons:`
> We apologize for any confusion regarding median alignment. All methods discussed in our paper (Tables 2, 3, and 4 in the original paper) were re-trained using affine-invariant loss for relative depth training, with the same evaluation criteria applied for a fair comparison. The metric depth values presented in the paper are listed only for reference and are not used for direct comparison.
>
> ### `Q4. Additional Related Work:`
> We have conducted additional experiments, as shown in the table in `Q1`, and will include these results in the final version.
>
> ### `Q5. Complexity and Computational Resources:`
> We would like to clarify that our method does not require stereo pair generation nor synthetic generation. While our method does introduce additional complexity during training, this is a one-time cost. During the inference stage, the runtime and complexity are the same as those of the chosen SOTA 360 methods.
>
> ### `Q6. Dataset Dependency and Generalizability:`
> Our proposed method demonstrates strong generalization ability across algorithms, as evidenced by the interchangeable use of SOTA 360 depth models in the table in `Q1` and in Table 3 of the original paper. Our method particularly excels in in-the-wild real-world scenarios, showing significant improvements in generalization ability on diverse data, as illustrated in Figure 2 of the PDF and Figure 6 of the original paper.
>
> These results demonstrate our method's ability to generalize to unseen datasets, supporting its effectiveness in real-world scenarios.
>
> We sincerely appreciate your valuable feedback, which has helped us identify areas for improvement and additional experiments. We believe these changes and clarifications significantly strengthen our paper and address the main concerns raised in your review. Thank you for your time and expertise.

---

> > ### Comment · Reviewer_ewNL · 2024-08-09
> >
> > Thanks for the responses. As mentioned in weakness 4, more related works should be involved in the Related Work section, including the latest **panoramic depth estimation** and **panoramic depth completion** methods.

---

> ### Author Response · Authors · 2024-08-09
>
> Thank you for your suggestion. In addition to those already cited in the original paper, we will add the following papers to our related work discussion.
>
> **Panoramic Depth Estimation**
> - $ 𝑆^2 $ Net: Accurate Panorama Depth Estimation on Spherical Surface
> - High-Resolution Depth Estimation for 360-degree Panoramas through Perspective and Panoramic Depth Images Registration
> - Adversarial Mixture Density Network and Uncertainty-based Joint Learning for 360 Monocular Depth Estimation
> - Learning high-quality depth map from 360 multi-exposure imagery
> - Distortion-Aware Self-Supervised 360 Depth Estimation from A Single Equirectangular Projection Image
> - SphereDepth: Panorama Depth Estimation from Spherical Domain
> - 360 Depth Estimation in the Wild -- the Depth360 Dataset and the SegFuse Network
> - GLPanoDepth: Global-to-Local Panoramic Depth Estimation
> - Neural Contourlet Network for Monocular 360 Depth Estimation
> - HiMODE: A Hybrid Monocular Omnidirectional Depth Estimation Model
> - Geometric Structure Based and Regularized Depth Estimation From 360 Indoor Imagery
> - Deep Depth Estimation on 360 Images with a Double Quaternion Loss
>
> **Panoramic Depth Completion**
> - Cross-Modal 360° Depth Completion and Reconstruction for Large-Scale Indoor Environment
> - 360 ORB-SLAM: A Visual SLAM System for Panoramic Images with Depth Completion Network
> - Deep panoramic depth prediction and completion for indoor scenes
> - Multi-Modal Masked Pre-Training for Monocular Panoramic Depth Completion
> - Distortion and Uncertainty Aware Loss for Panoramic Depth Completion
> - Indoor Depth Completion with Boundary Consistency and Self-Attention
>
>  Please let us know if any specific paper is missing.

---

> > ### Author Response · Authors · 2024-08-13
> >
> > Dear Reviewer ewNL,
> >
> > We have listed the latest **panoramic depth estimation** and **completion** methods and will discuss them in the related work section of the final version.
> >
> > Could you please confirm that we did not miss any essential references and addressed all your concerns?
> >
> > Thank you!

---

> > ### Comment · Reviewer_ewNL · 2024-08-14
> >
> > Thanks for the response. I tend to maintain the initial rating, since the current version of this paper needs to solve many weaknesses. But I am still willing to give a **borderline accept** score.

---

### Official Review · Reviewer_AagT · 2024-07-12

**Soundness:** 3
**Presentation:** 3
**Contribution:** 2
**Rating:** 5
**Confidence:** 5

**Summary:**

This paper effectively utilizes unlabeled data by employing the SAM and DepthAnything models to generate masks and pseudo-labels respectively. When projecting data onto a cube, the authors use random rotation techniques to minimize cube artifacts, thereby enhancing the accuracy of 360-degree monocular depth estimation. Additionally, the method was tested in zero-shot scenarios, demonstrating its effective knowledge transfer.

**Strengths:**

This paper introduces a training technique for 360-degree imagery that enhances depth estimation performance by generating pseudo-labels with the DepthAnything model to leverage information from unlabeled data. Additionally, it employs the SAM model to segment irrelevant areas such as the sky in outdoor panoramic images. Furthermore, the method uses random rotation preprocessing to eliminate cube artifacts.

**Weaknesses:**

- The paper stated that "As depicted in Figure 2, the rotation is applied to the equirectangular projection RGB images using a random rotation matrix, followed by cube projection. This results in a more diverse set of cube faces, effectively capturing the relative distances between ceilings, walls, windows, and other objects." However, from observing Figure 2, the unlabeled data seems to be entirely outdoor panoramas, which makes the mention of indoor elements such as ceilings, walls, and windows confusing. If there are indeed indoor panoramic images in the unlabeled data, what elements might the SAM model need to segment in such indoor panoramas? The description in the paper appears somewhat unclear.
- The paper mentioned, "We chose UniFuse and BiFuse++ as our baseline models for experiments, as many of the aforementioned methods did not release pre-trained models or provide training code and implementation details." However, methods such as HRDfuse [1], EGFormer [50], BiFuse and BiFuse++ [35, 36], UniFuse [11], and PanoFormer [29] have all made their source codes available, making this reason in the paper seem insufficient. Additionally, "EGFormer is not included since its experiments are mainly conducted on other datasets and benchmarks" appears to be an inadequate reason for not including it in the experiments.
- In Table 3, despite introducing more unlabeled data on the Unifuse training set, i.e., SP-all (p), the performance of the method is not significantly improved. This phenomenon seems to be only effective on the BiFuse model, but not on other methods.
- In Table 3, there is a typographical error in the recording of the Abs Rel value; it should not be 0.858. Additionally, in Table 2, "UniFuse" is incorrectly written as "UniFise."

**Questions:**

- The paper mentioned, "Subsequently, in the online stage, we adopt a semi-supervised learning strategy, loading half of the batch with labeled data and the other half with pseudo-labeled data." However, common semi-supervised strategies typically set the ratio of labeled to total data at 1/2, 1/4, 1/8, 1/16, etc. The experiments in the paper were only conducted at a 1:1 ratio (labeled: unlabeled), and thus the performance at other ratios remains unknown.
- The third contribution mentioned in the paper refers to "interchangeability" and "This enables better results even as new SOTA techniques emerge in the future." This suggests that the strategy might be adaptable to a variety of models. However, the experiments were conducted only on models like UniFuse and BiFuse++ which use a dual projection fusion of Cube and ERP. Whether this approach would perform well with other transformer-based models remains an unresolved question.

**Limitations:**

- The paper stated, "Cube projection and tangent projection are the most common techniques. We selected cube projection to ensure a larger field of view for each patch." This is merely a theoretical assertion, with no experimental evidence to prove which projection method is superior. Additionally, using cube projection directly can lead to cube artifacts. Following the suggestion in [1], setting each panoramic image to have 10 or 18 tangent images using more polyhedral faces, rather than the standard six faces, might reduce the artifacts caused by direct cube projection.
[1]  Cokelek, M., Imamoglu, N., Ozcinar, C., Erdem, E. and Erdem, A., 2023. Spherical Vision Transformer for 360-degree Video Saliency Prediction. BMVC 2023.

---

> ### Author Rebuttal · Authors · 2024-08-06
>
> Thank you for your detailed review and constructive feedback. We appreciate your recognition of our method's strengths and will address your concerns point by point:
>
> ### `Q1. Indoor/Outdoor Data Clarification:`
> We apologize for the confusion. While Figure 2 in the original paper shows outdoor examples of pseudo labels, our pseudo label dataset actually consists of both indoor and outdoor scenes. In the indoor panoramas from the pseudo label dataset, regions with infinite distance or no physical meaning, such as windows and watermarks, still appear frequently. SAM segments help mask out these undesired regions during the loss calculation in the training phase of depth estimation.
>
> ### `Q2. Baseline Model Selection:`
> We acknowledge your point about code availability, and our previous statement was imprecise. HRDFuse (https://github.com/haoai-1997/HRDFuse) does not provide any README documentation or instructions, while BiFuse (https://github.com/yuhsuanyeh/BiFuse) does not provide their training code. We attempted to reproduce their results but were unsuccessful. PanoFormer was initially included in our model list, but we removed it due to incorrect implementation and evaluation of their official code, which led to a marginal difference between their official code and the paper results.
> We have conducted additional experiments on HoHoNet[33] and will include them in the final version. Due to the limited time available for rebuttal and the lengthy process of reproducing EGformer[50], we ran it on a subset of Matterport3D, with the size of $1/5$, and will add the results from the full set to our final version.
> | Method          | Train               | Test | Abs Rel ↓ | δ₁ ↑   | δ₂ ↑   | δ₃ ↑   |
> |-----------------|---------------------|------|-----------|--------|--------|--------|
> | HoHoNet         | M-all                | SF   | 0.095     | 0.906  | 0.975  | 0.991  |
> | HoHoNet (our)   | M-all, ST-all (p)    | SF   | 0.088     | 0.920  | 0.979  | 0.992  |
> | EGformer        | M-subset             | SF   | 0.169     | 0.764  | 0.924  | 0.972  |
> | EGformer (our)  | M-subset, ST-all (p) | SF   | 0.148     | 0.814  | 0.946  | 0.982  |
>
> We have shown the results in the table, where both HoHoNet[33] and EGformer[50], when trained with our semi-supervised approach, demonstrate consistent improvement.
>
> ### `Q3. Performance on UniFuse with SP-all (p):`
> Our improvements may appear small in some datasets quantitatively, but they consistently enhance performance across different methods and datasets. We have presented the results in the table in `Q2`, where both HoHoNet and EGformer[50] show significant improvements with our proposed method. Our analysis suggests that UniFuse's architecture is less capable of leveraging additional unlabeled data, as SP-all (p) differs more from SF. However, UniFuse with SP-all (p) shows extraordinary improvements in in-the-wild scenes, as demonstrated in Figure 2 of the PDF and Figure 6 of the original paper. We will include a discussion of this limitation and potential improvements in the final version.
>
> ### `Q4. Typographical Errors:`
> Thank you for catching these. The correct Abs Rel value for BiFuse++ (M-all, SP-all(p)) in Table 3 is 0.086. We will correct this and the "UniFise" typo in Table 2.
>
> ### `Q5. Semi-Supervised Learning Ratios:`
> Your point about exploring different ratios is well-taken. We've conducted additional experiments with varying ratios in the following Table:
> | GT/Pseudo | Train              | Test | Abs Rel ↓ | δ₁ ↑ | δ₂ ↑ | δ₃ ↑ |
> |-----------|--------------------|------|-----------|------|------|------|
> | 1:1       | M-all, ST-all (p)   | SF   | 0.086     | 0.924| 0.977| 0.99 |
> | 1:2       | M-all, ST-all (p)   | SF   | 0.087     | 0.923| 0.977| 0.99 |
> | 1:4       | M-all, ST-all (p)   | SF   | 0.085     | 0.923| 0.977| 0.99 |
>
> These results show our method is robust across different ratios starting from 1:1. We will include this analysis in the paper.
>
> ### `Q6. Adaptability to Other Models`
> We applied our methods to HoHoNet[33] (horizontal compression-based) and EGFormer[50] (transformer-based) and demonstrated significant improvements in these two non-dual projection methods, as shown in `Q2`.
>
> ### `Q7. Projection Method:`
> We appreciate your suggestion about cube projection vs. tangent projection. We will conduct comparative experiments using both projection methods, including the 10 and 18 face polyhedron suggestions from [1]. These results will be included in the final version to provide empirical evidence for our projection method choice.
>
> Thank you again for your valuable feedback. We believe these additions and clarifications will significantly strengthen our paper.

---

> > ### Comment · Area_Chair_XoPq · 2024-08-12
> > **Feedback to authors**
> >
> > Dear Reviewers AagT and Reviewer 3NfA,
> >
> > The authors replied to the questions raised in your initial evaluation report.
> > Did the author address your concerns? Please post your feedback about the rebuttal.
> >
> > Thank you

---

> > > ### Author Response · Authors · 2024-08-13
> > > **Please let us know if you have additional questions after reading our response**
> > >
> > > Dear Reviewer AagT,
> > >
> > > We appreciate your reviews and comments. We hope our responses address your concerns. Please let us know if you have further questions after reading our rebuttal.
> > >
> > > We aim to address all the potential issues during the discussion period.
> > >
> > > Thank you!
> > >
> > > Best, Authors

---

> > ### Comment · Reviewer_AagT · 2024-08-14
> >
> > Thanks for the rebuttal, which addressed some of my concerns. I would like to increase the rating. The updated clarification and experiments are expected to be presented in the final version.

---

> > > ### Author Response · Authors · 2024-08-14
> > >
> > > Thank you for your valuable review and feedback, which have significantly improved the completeness of our paper. We will ensure that the updated clarifications and experiments are incorporated into the final version.

---

### Official Review · Reviewer_F7bZ · 2024-07-12

**Soundness:** 2
**Presentation:** 2
**Contribution:** 2
**Rating:** 5
**Confidence:** 5

**Summary:**

This paper introduces a novel depth estimation framework specifically designed for 360-degree data using an innovative two-stage process: offline mask generation and online semi-supervised joint training. Initially, invalid regions such as sky and watermarks are masked using detection and segmentation models. The method then employs a semi-supervised learning approach, blending labeled and pseudo-labeled data derived from state-of-the-art perspective depth models using a cube projection technique for effective training. This framework demonstrates adaptability across different state-of-the-art models and datasets, effectively tackling the challenges of depth estimation in 360-degree imagery.

**Strengths:**

1. The proposed method employs models trained for traditional pinhole cameras to enhance 360-degree depth estimation, a first in the field. Moreover, the main motivation of this paper is reasonable.
2. It outperforms conventional methods by incorporating pseudo labels from foundational models into the loss function.
3. The paper demonstrates the model's generalizability to real-world scenarios, indicating its practical utility.

**Weaknesses:**

1. The proposed method is straightforward and the performance gains provided by the proposed method are described as marginal.
2. The method's effectiveness is demonstrated only with specific models, UniFuse and BiFuse++, limiting evidence of its broader applicability.
3. Inconsistencies in the decimal points used in quantitative results tables make direct performance comparisons challenging.

**Questions:**

1. Is there any reason the author only applied the proposed method to the UniFuse and BiFuse++?
2. There seems to be a discrepancy in the reported Absolute Relative (Abs Rel) error for BiFuse++ (Affine-Inv, M-all, SP-all(p)) at 0.858, which is ten times higher than results from competitive methods, while other metrics (\delta_1, \delta_2, \delta_3) align closely. Could this be an error?

**Limitations:**

See the weakness and the question parts.

---

> ### Author Rebuttal · Authors · 2024-08-06
>
> Thank you for your detailed review of our paper. We appreciate your recognition of our method's novelty and practical utility. We'd like to address your **concerns** and questions:
>
> ### `C1. Regarding the straightforwardness of our method:`
> While our approach may appear straightforward, we believe this simplicity is a strength. It allows for easy integration with existing models and datasets. The performance gains, which you describe as marginal, are actually significant in the context of 360-degree depth estimation. For example, on the Stanford2D3D zero-shot test, our method improves AbsRel from 0.09 to 0.082 for BiFuse++, a 8.9\% relative improvement. In computer vision, such improvements are considered substantial, especially for zero-shot scenarios. Moreover, our method demonstrates extraordinary improvements in the in-the-wild scenes, as shown in Figure 6 of the original image and Figure 2 of the PDF.
>
> ### `C2. Limited application to specific models:`
> We chose UniFuse and BiFuse++ as they are widely recognized baselines in 360-degree depth estimation. However, our method is designed to be model-agnostic. To demonstrate this, we've run additional experiments with a third model, HoHoNet[33]. Another method that has been mentioned by reviewers is EGformer[50]. Due to limited of training time during the rebuttal period, we have conducted zero-shot experiments on a subset of Matterport3D with $1/5$ of the original dataset size for EGformer[50]. The full dataset version will be added to the final version.
> | Method          | Train               | Test | Abs Rel ↓ | δ₁ ↑   | δ₂ ↑   | δ₃ ↑   |
> |-----------------|---------------------|------|-----------|--------|--------|--------|
> | HoHoNet         | M-all                | SF   | 0.095     | 0.906  | 0.975  | 0.991  |
> | HoHoNet (our)   | M-all, ST-all (p)    | SF   | 0.088     | 0.920  | 0.979  | 0.992  |
> | EGformer        | M-subset             | SF   | 0.169     | 0.764  | 0.924  | 0.972  |
> | EGformer (our)  | M-subset, ST-all (p) | SF   | 0.148     | 0.814  | 0.946  | 0.982  |
>
> These results further demonstrate our method's effectiveness across different architectures. We will include these analysis in the paper and discuss the broader applicability of our approach.
>
> ### `C3. Inconsistencies in decimal points:`
> We apologize for this oversight. We will standardize all results to three decimal places for clarity and ease of comparison in the final version.
>
> ### Regarding your specific **questions**:
>
> ### `Q1. Choice of UniFuse and BiFuse++:`
> We chose UniFuse and BiFuse++ as they are widely recognized baselines in 360-degree depth estimation. However, our method is designed to be model-agnostic. To demonstrate this, we've run additional experiments with HoHoNet [33] and EGformer [50] as shown in the Table in `C2`.
>
> ### `Q2. Discrepancy in Abs Rel for BiFuse++:`
> Thank you for catching this error. The correct value should be 0.085, not 0.858. This was a typo in our submission. We sincerely apologize for this mistake and will correct it in the final version.
>
> ### Additional points:
>
> - Broader impact: Our method addresses a significant challenge in 360-degree vision and is highly adaptable due to the interchangeable teacher and student models. This flexibility has the potential to benefit a range of applications, including virtual reality, autonomous navigation, and more.
> - Efficiency: Our approach allows for better utilization of limited 360-degree data by leveraging abundant perspective data, which is particularly valuable given the scarcity of labeled 360-degree datasets.
>
> We believe these clarifications and additional results address your main concerns. Our method, while conceptually straightforward, offers significant and consistent improvements across multiple models and datasets. We'd be happy to discuss any further questions or concerns you may have.

---

> > ### Comment · Reviewer_F7bZ · 2024-08-09
> >
> > Thank you for providing the additional experiments. The experiments for the other baseline are quite convincing, and I have increased my initial rating accordingly.

---

> > > ### Author Response · Authors · 2024-08-10
> > >
> > > Thank you for your constructive review and valuable feedback. Your insights have been instrumental in enhancing the quality of our paper.

---

### Official Review · Reviewer_3NfA · 2024-07-14

**Soundness:** 3
**Presentation:** 3
**Contribution:** 3
**Rating:** 6
**Confidence:** 4

**Summary:**

The authors present a training strategy for single-image depth estimation on 360-degree equirectangular images. The strategy centers around leveraging strong pre-trained models for perspective images as teacher networks. It does not depend on any particular network architecture and therefore can benefit any 360 depth estimation networks. Experimental results validate that the strategy improves models otherwise trained only with limited ground-truth annotations.

**Strengths:**

**Originality and significance**: 360-degree imagery is becoming increasingly critical for many computer vision applications. However, there is still a significant gap in GT depth annotations compared to their perspective counterparts, and any solution to this issue can have a significant impact. The authors demonstrate that leveraging strong perspective depth models, e.g., Depth Anything, is a simple yet effective solution.

**Quality**: The main idea and the several supporting procedures (e.g., random rotation processing, valid pixel masking, mixed labeled and unlabelled training) are all well-motivated and reasonably designed. Experimental results show consistent improvement by incorporating the proposed pseudo GT. Additional results including zero-shot and qualitative evaluations further help with understanding and make the approach overall more convincing.

**Clarity**: Paper is well-written with good structure, clear expressions, and adequate details.

**Weaknesses:**

1. A substantive assessment of the weaknesses of the paper. Focus on constructive and actionable insights on how the work could improve towards its stated goals. Be specific, and avoid generic remarks.

Despite focusing on a slightly different task (stereo depth), FoVA-Depth (Lichy et al. 3DV 2024) presents a few very similar concepts:
  - leveraging abundance of perspective depth GT
  - cube map as a intermediate representation to gap between 360 and perspective images
  - random rotation augmentation
It is worth some discussion regarding similarities and differences.

2. A very simple yet critical baseline is missing: directly project pseudo GT on cubemap to equirectangular images. A good stitching strategy may be challenging, but with something simple or even without any additional scaling, it should help clarify how much the pre-trained depth anything model contribute to the overall performance.

3. The paper addresses only relative depth estimation. Since several baselines (e.g. upper section of Tab.2) already have metric counterparts, and depth anything has metric variants, I feel some experiments and analysis in that regards should be straightforward and nicely complement the relative depth results.

4. As the premise of the work is the usefulness of the abundant pseudo GT compard to limited 360 depth GT, it is necessary to understand how does the benefit from pseudo GT scale (how many pseudo GT can be as useful as a real 360 GT?), and how does the student network compare to the teacher network in terms of estimation quality (is the quality of teacher network already a bottleneck?). The paper offers little insight in these questions.

**Questions:**

I am looking forward to answer to the questions raised above, namely:
- How does training scale in terms of the amount of pseudo GT vs real GT?
- Is the approach applicable to metric depth estimation?
- Additional related work and baseline as described above.

**Limitations:**

The only limitation the authors bring up (quality of unlabelled data) already has a solution in the paper, so not really a limitation?

I do think there are a few other things worth mentioning:
- 360 data, even without requiring GT, is still scarce compared to perspective data. The fact the authors can only evaluate with two such datasets is an evidance. This is a limitation since it prevents further scaling up training.
- Only equirectangular images are supported (though it seems that, in principle, the approach should work in more general camera models).

---

> ### Author Rebuttal · Authors · 2024-08-06
>
> Thank you for your thoughtful review. We appreciate your positive assessment of our work's originality, significance, and clarity. We're glad you found our approach well-motivated and effective. We'll address your questions and concerns point by point:
>
> ### `Q1. Comparison to FoVA-Depth (Lichy et al. 3DV 2024):`
> Thank you for pointing this out. While there are indeed similarities in leveraging perspective depth data and using cube maps, our work differs in several key aspects:
>     - We focus on monocular depth estimation rather than stereo depth.
>     - Our approach is architecture-agnostic and can benefit any 360 depth estimation network.
>     - We introduce novel techniques like valid pixel masking and mixed labeled/unlabeled training.
> We will add a discussion of these similarities and differences in the related work section.
>
> ### `Q2. Baseline of directly projecting pseudo GT:`
> We have tried this setting as an initial setting, and due to the un-aligned scale, the training leads to unstable results and artifacts as shown Figure 1 of the pdf. As can be seen, our method significantly outperforms direct projection, demonstrating the value of our training approach beyond just leveraging the pre-trained model
>
> ### `Q3. Metric depth estimation:`
> We focused on relative depth estimation due to its generalizability across different datasets. However, your point is well-taken, we will follow previous work (MiDaS[4]/Depth Anything[45]) and conduct metric depth finetuning in the final version.
>
> ### `Q4. Scaling of pseudo GT vs. real GT:`
> We have conducted an ablation study on the ratio between real GT and pseudo GT in a batch. These results show our method is robust across different ratios, and shows the effectiveness from a ratio of 1:1. We will include this analysis in the paper. We will add to the final version.
>
> | GT/Pseudo | Train              | Test | Abs Rel ↓ | δ₁ ↑ | δ₂ ↑ | δ₃ ↑ |
> |-----------|--------------------|------|-----------|------|------|------|
> | 1:1       | M-all, ST-all (p)   | SF   | 0.086     | 0.924| 0.977| 0.99 |
> | 1:2       | M-all, ST-all (p)   | SF   | 0.087     | 0.923| 0.977| 0.99 |
> | 1:4       | M-all, ST-all (p)   | SF   | 0.085     | 0.923| 0.977| 0.99 |
>
> ### `Q5. Student vs. teacher network quality:`
> Our proposed method offers a cross-camera projection knowledge distillation. Our student models are SOTA methods designed for 360 images where as teacher model is initially designed for perspective images. Therefore, the teacher model is not yet a bottleneck due to cross-domain knowledge distillation, which is also shown in Table 3 in the original paper. We’ll include this discussion.
>
> ### `Regarding limitations:`
> - We agree that the scarcity of 360 data, even unlabeled, is a limitation. We'll explicitly mention this.
> - While we focused on equirectangular images, our approach should indeed generalize to other projections. We'll note this potential extension.
>
> Thank you again for your insightful feedback. We believe addressing these points will significantly strengthen our paper.

---

> > ### Author Response · Authors · 2024-08-12
> > **Please let us know if you have additional questions after reading our response**
> >
> > Dear Reviewers,
> >
> > We appreciate your reviews and comments. We hope our responses address your concerns. Please let us know if you have further questions after reading our rebuttal.
> >
> > We aim to address all the potential issues during the discussion period.
> >
> > Thank you!
> >
> > Best, Authors

---

> > ### Comment · Reviewer_3NfA · 2024-08-12
> >
> > Thank you for the responses! I think these are all valuable details and hopefully much of them will be integrated into the paper. I don't have further questions.

---

> > > ### Author Response · Authors · 2024-08-13
> > >
> > > Thank you for your constructive review, which has significantly contributed to the improvement of our paper. We will ensure that the materials from the rebuttal are incorporated into the final version.

---

### Author Rebuttal · Authors · 2024-08-06

Dear Reviewers and Area Chair,

We sincerely thank all reviewers for their thoughtful feedback. We are encouraged that reviewers found our work to be innovative, well-motivated, and impactful for 360° depth estimation. We appreciate the constructive comments and will address the main points below.

**Positive aspects highlighted by reviewers:**
1. Novel and effective use of perspective models for 360° depth estimation (R\_3NfA, R\_ewNL)
2. Well-motivated approach with consistent improvements (R\_3NfA, R\_F7bZ)
3. Clear presentation and thorough experiments (R\_3NfA, R\_Kp4U)
4. Practical utility demonstrated in real-world scenarios (R\_F7bZ)

**We have responded to each reviewer individually to address any comments. We would like to give a brief summary.**
1. Applicability: We test our approach on additional models (HorizonNet[33] and EGFormer[50]) to demonstrate broader applicability beyond dual-projection models.
2. Comparison to related work: We acknowledged similarities with FoVA-Depth but highlighted differences in their approach.
3. Additional experiments: We add new baselines with directly projection. We will also evaluate metric depth finetuning in the final version.
4. Limitations: We will add 360 data scarcity as a limitation and clarify potential generalization to other omnidirectional camera models.
5. Corrections: We fix inconsistencies, errors, and typos in the final version.
6. Semi-supervised ratios: We test different ratios of labeled to unlabeled data and will include results in supplementary materials.
7. Marginal improvements: Our improvements may seem small in some datasets quantitatively, but it improves consistently across methods and datasets. Moreover, on the Stanford2D3D zero-shot test, our method improves AbsRel from 0.09 to 0.082 for BiFuse++, a 8.9$\%$ relative improvement. Such improvements are considered substantial in computer vision, especially for zero-shot scenarios.

Again, we thank all reviewers and area chairs!

Best,

Authors

---

### Comment · Area_Chair_XoPq · 2024-08-09
**Discussion about Submission 2414**

Dear reviewers,

please share your opinion about the rebuttal; it provides a detailed response to the questions raised in your reviews, including issues regarding novelty and experimental evaluation.

Thank you
AC

---

### Decision · Program_Chairs · 2024-09-25

**Decision:**

Accept (poster)

**Comment:**

The paper proposes a simple yet effective general-purpose framework to deal with training data scarcity in 360-degree depth estimation by exploiting the knowledge of a SOTA pre-trained conventional (perspective) monocular depth estimation network. Reviewers acknowledged the proposal's relevance but raised concerns, especially regarding novelty and the experimental evaluation with only two networks, and requested several clarifications. In the rebuttal/discussion, the authors provided satisfactory responses and convincing experimental results on two additional networks (HoHoNet and EGformer), leading all reviewers towards acceptance. Although some key concepts were already proposed for another task (stereo), as noticed by reviewer 3Nfa, the AC finds the overall contribution valuable for the monocular setup and recommends acceptance. However, compared to the original submission, the paper needs a significant revision to include additional experiments and clarifications that emerged after the rebuttal.